# Natural Contaminants in Wines: Determination of Biogenic Amines by Chromatographic Techniques

**DOI:** 10.3390/ijerph181910159

**Published:** 2021-09-27

**Authors:** Giuliana Vinci, Lucia Maddaloni, Sabrina A. Prencipe, Roberto Ruggieri

**Affiliations:** Department of Management, Sapienza University of Rome, Via del Castro Laurenziano 9, 00161 Rome, Italy; lucia.maddaloni@uniroma1.it (L.M.); sabrinaantonia.prencipe@uniroma1.it (S.A.P.); roberto.ruggieri@uniroma1.it (R.R.)

**Keywords:** contaminants, red wines, white wines, food quality, biogenic amines, winemaking processes, food safety, microorganisms, alcoholic fermentation, malolactic fermentation

## Abstract

Biogenic amines (BAs) are natural contaminants of wine that originate from decarboxylase microorganisms involved in fermentation processes. The primary relevance of biogenic amines in food could have both toxic effects on consumers’ health (i.e., allergic reactions, nausea, tremors, etc.), if present at high concentrations, and concurrently it can be considered as a remarkable indicator of quality and/or freshness. Therefore, the presence of nine biogenic amines [Tryptamine (TRP), ß-phenylethylamine (ß-PEA), putrescine (PUT), cadaverine (CAD), histamine (HIS), serotonin (SER), tyramine (TYR), spermidine (SPD), and spermine (SPM)] was investigated in red and white wine samples, which differed in the winemaking processes. The qualitative-quantitative determination of BAs was carried out by chromatographic methods (HPLC-UV/Vis and LC-ESI-MS). The analysis showed that both winemaking processes had all the nine BAs considered in the study at different amounts. Data showed that red wines had a higher concentration of PUT (10.52 mg L^−1^), TYR (7.57 mg L^−1^), and HIS (6.5 mg L^−1^), the BAs most involved in food poisoning, compared to white wines, probably related to the different type of fermentation (alcoholic and malolactic).

## 1. Introduction

Biogenic amines (BAs) are a class of organic, basic, and low-molecular weight compounds with heterocyclic (histamine, tryptamine), aliphatic (spermine, spermidine, putrescine and cadaverine) and aromatic (tyramine, phenylethylamine) structures [1,2]. They can be endogenous or exogenous in plants, as well as in animal and microorganisms, where they play, at lower concentration, an important role in physiological and metabolic functions—e.g., membrane stabilization, nucleic acid regulation, and protein synthesis [3]. The most amines occurring in food originate from proteolytic processes that make available large quantities of amino acids, which are the ideal substrate for enzymatic decarboxylation reactions. In addition, BAs may also be synthetized from the amination and transamination of aldehydes and ketones by the amino-acetic transaminases. BAs can be synthesized both in perishable and fresh food (i.e., fruits and vegetables, meat, fish, etc.), which are exposed to decarboxylase-positive microorganisms [4], as well as in fermented and/or processed food (i.e., wine, beer, coffee, chocolate, etc.), as a direct consequence of their transformation process—e.g., alcoholic and malolactic fermentation. The primary relevance of biogenic amines in food could have both toxic effects on consumers’ health, if present at high concentrations, and concurrently it can be considered as a remarkable indicator of quality and/or freshness. From the toxicological point of view, BAs have been widely investigated as human harmful compounds, since their excessive food-mediated intake and a reduced or absent catabolism may induce symptoms that are similar to those of food poisoning: migraine headaches, gastric disorders, nausea, cardiac palpitations, and psychoactive effects [4,5]. The toxic effect most attributable to the ingestion of BAs is scombroid syndrome (Scombrotoxin Fish Poisoning, SFP), also called “Histamine poisoning”, as this amine is the one responsible for this intoxication. However, the other BAs enhance the effects of histamine. Therefore, the toxicity of BAs in food or beverages is mainly due to a synergistic effect of several BAs [5]. Table 1 shows the physiological and pathological effects of BAs on human health. Below physiological conditions, BAs can be metabolized by three different enzymes, present in the gastrointestinal tract [diamino oxidase (DAO), monoamine oxidase (MAO), and histamine-N-methyltransferase (HMT)], which have the function of inactivating BAs by oxidizing the amino groups. Through these three enzymes, the human body has the ability to inactivate BAs that are normally taken with food and beverages [4,6]. Furthermore, in alcoholic beverages, it has been shown that the presence of ethanol and acetaldehyde, its catabolite, inhibits the enzymatic activity of BAs detoxifying enzymes (DAO, MAO, and HMT) and increases the biogenic amines permeability in the gastrointestinal wall, consequently increasing its toxic effects. This aspect is of relevant importance in alcoholic beverages, such as wine, which are eaten on its own or paired with other foods containing high concentrations of BAs (i.e., cheese, fish, meat, etc.) [7].

However, the toxic effects of BAs are dose-dependent, and the severity of the toxicity response is also influenced by personal sensitivity to these compounds. The symptoms, similar to those of food poisoning (nausea, vomiting, respiratory dysfunctions, itching, skin rash, etc.), have a variable duration between 8–12 h and they can therefore occur with more or less serious consequences even in subjects with a correct functioning of the enzymatic activity [4,8]. The daily dose of BAs acceptable to the human body is not yet known, as the toxic effects of the individual BAs are correlated and enhanced by their co-presence in food. Based on this, the EFSA (European Food Safety Authority) has defined that the dose, referred to histamine, for which the human body is able to activate the defense mechanisms is equal to about 25–50 mg and that the poisoning occurs following the intake of BAs equal to about 70–300 mg.

However, although their potential toxicity is known, to date, there is no legislation that allows for limiting the sale of products with high BA content. Currently, European legislation regulates only the presence of histamine in fish and fishery products (Reg. 2073/2005), while, as regards wine, only some European Countries (Germany, France, The Netherlands, Belgium, and Austria), arbitrarily, they proposed limits for histamine, ranging from 2 to 10 mg L^−1^ [5].

The formation of BAs therefore presupposes the co-presence of three factors: (i) a precursor, that is a specific amino acid for each specific BAs; (ii) contaminating microorganisms with decarboxylase activity; (iii) favorable environmental conditions (i.e., pH, temperature, water activity, etc.). Wine is an excellent substrate for BAs synthesis while ensuring the presence of amino acids, microbial populations with decarboxylating activity and a generally favorable environment for microorganism growth. Therefore, the BAs determination in wines is not only important to safeguard the consumers’ health, but also for food quality assessment, because the presence of biogenic amines may influence the organoleptic characteristics of the finished product. The winemaking processes (Figure 1), which occur following different biochemical and metabolic pathways by microorganisms (bacteria, fungi, and yeasts), can lead to the formation of BAs [2,6].

The presence of these compounds in wine occurs at different points in the winemaking process. Especially, it can be influenced by the conservation conditions of grapes bunches, by their degree of ripeness and by the pedoclimatic conditions in which the wine is cultivated. The main step involving the BAs formation is the fermentation phase, in fact, depending on the type of microorganism involved and the type of fermentation (alcoholic and malolactic), not only the concentration but also the type of BAs present can be influenced in the finished product [1]. Table 2 shows the main microorganisms involved and the main BAs synthesized for the two types of fermentation.

Two different winemaking processes, red and white ones, were taken into consideration as they are the most representative wine classes of the Italian market. In this study, to evaluate safety and quality of wines samples, nine biogenic amines were analyzed (TRP, HIS, TYR, β-PEA, CAD, PUT, SER, SPD, and SPM) and the BA profile was studied to identify the difference between red and white wines. The content of BAs has been analyzed by two chromatographic methods (HPLC UV/Vis and LC-ESI-MS).

## 2. Materials and Methods

### 2.1. Chemicals

Tryptamine (TRP), ß-phenylethylamine (ß-PEA), putrescine (PUT), cadaverine (CAD), histamine (HIS), serotonin (SER), tyramine (TYR), spermidine (SPD), and spermine (SPM) with a purity of 99%, the internal standard 1.7 diaminoheptane (IS), derivatizying agent (dansyl chloride) and heptafluorobutyric acid (HFBA) were purchased from Sigma-Aldrich (St. Louis, MO, USA). The sodium hydroxide (NaOH) and sodium carbonate (Na_2_CO_3_) used for derivatization were purchased by Sigma-Adrich (Milan, Italy). The following solutions were used for the sample preparation and the chromatographic determination of the compounds: Acetonitrile (HPLC grade, Merck, Darmstadt, Germany), Methanol (Carlo Erba Reagenti, Milan, Italy), Perchloric acid (HClO_4_) 65% (Merck, Darmstadt, Germany), Ammonium hydroxide (NH_4_OH) 25% (Carlo Erba Reagenti), distilled water purified using a Milli-Q system (Millipore, Bedford, MA, USA) and ultrapure water (Millipore, USA).

### 2.2. Instruments

The following instruments were used: Sartorius model 1712 analytical balance, ALC 4236 centrifuge, homogenizer Universal Laboratory Aid MPW-309, Bandelin Sonorex RK100H water and ultrasonic thermostatic bath, FALC model F60 magnetic stirrer, Whatman 0.20 µm 100 (PTFE) syringe filters, Sigma Aldrich (Milan, Italy). Chromatographic analysis was performed using an ATVP LC-10 HPV binary pump with an SPD-10AVP UV detector (Shimadzu, Kyoto, Japan) operating to λ = 254 nm. A Supelcosil LC-18 column (250 mm × 4.6 mm, 5 µm) with a Supelguard LC-18 (Supelco, Bellefonte, PA, USA) pre-column was used for the determination of BAs. The following instrumentation was used for the analysis with the LC-ESI-MS system: Thermoquest (Manchester, UK) model P2000 with Alltima column (Alltech, IL, USA) C18 in reverse phase (250 mm × 4.6 mm id, dimension of particles 5 μm). Regarding mass spectrometry, the Finnigan AQA single quadrupole bench-top mass spectrometer was used. Instrument control, data acquisition and processing were carried out with Mass Lab (version 2.22) of Thermoquest Finnigan (Manchester, UK).

### 2.3. Standard Solution

For each BAs (TRP, ß-PEA, PUT, CAD, HIS, SER, TYR, SPD and SPM), individual standard solutions were prepared at 1 mg mL^−1^ in purified water and kept in the dark at 4 ± 1 °C. In addition, the standard solution containing all nine BAs (MIX 9) was obtained with 1 mL of each standard solution of the individual BAs and diluted in 25 mL with purified water. Different aliquots of these standard solutions were applied to obtain the standard solutions necessary for the construction of the calibration curves and for the execution of recovery experiments. The standard solutions were added with HFBA in order to obtain a final solution with an acid concentration of 5 mM. The concentrations of BAs injected for the construction of the calibration lines were 0.1, 0.4, 0.8, 4.0, 8.0, and 16.0 mg L^−1^. Furthermore, all solutions contained the IS, at the same concentration of 0.8 mg L^−1^.

### 2.4. Samples

Forty-four wine samples (24 white wines and 20 red wines) produced in different regions of Italy were purchased in local supermarkets. The wine samples were stored at room temperature and protected from light until the day of analysis. The wine samples were chosen to be as representative as possible of the Italian wine market and of the red and white winemaking processes.

### 2.5. Biogenic Amines Extraction

The extraction of biogenic amines from the wine samples was carried out by applying the method described in a previous article [37]. The wine samples were previously filtered using a 0.20 µm Millipore membrane filter. Subsequently, for HPLC-UV/Vis analysis, 10.3 M HClO_4_ was added to 25 mL of the filtered wine samples to obtain an acid solution at 0.2 M. Instead, for LC-ESI-MS analysis, HFBA was added to 25 mL of the filtered wine samples to obtain an acid solution at 5 mM. After a second filtration, an aliquot of 50 µL of the wine samples was injected into a chromatographic column.

### 2.6. HPLC-UV/Vis Method

Before performing the HPLC-UV/Vis analysis, the samples were subjected to derivatization, obtained through the use of dansyl-chloride [5-(dimethylamino) naphtalene1-sulfonyl chloride]. The derivatization reaction was carried out by adding 200 µL of 2N NaOH, 300 µL of saturated NaHCO_3_ solution, and 2 mL of dansyl-chloride solution (15 mg mL^−1^ in acetone) to 1 mL of extract. After stirring, the samples were left in the dark at room temperature for 20 min. To stop the reaction, 100 mL of 25% *v*/*v* NH_4_OH are added at the end and the final volume was brought to 5 mL with acetonitrile. The derivatized sample was subsequently filtered with 0.45 µm PTFE syringe filter. For the chromatographic determination of the BAs, a volume aliquot of 50 µL (loop 50 µL) was injected. Analyses were performed by use of a Supelcosil LC-18 column (150 mm × 4.6 mm, 5 µm), Supelco, Bellefonte, PA, USA) coupled with an UV detector (254 nm). The analyses were carried out maintaining a fixed temperature of 25 °C. The solvents used for the chromatographic separation were: (A) water purified and (B) acetonitrile. The elution program started with 3 min of isocratic elution (50% A; 50% B) reaching 10% A and 90% B after 20 min to finish with a further 3 min of isocratic elution. Finally, it took 4 min to restore the initial isocratic conditions (50% A 50% B). The flow was kept constant at 1.2 mL/min, for a total analysis time of 35 min.

### 2.7. LC-ESI-MS Method

During the LC-ESI-MS analyses, the ESI unit operated at 4.0 kV, the capillary was heated to 200 °C and as desolvation gas (300 L/h), as well as for the nebulizer (5 L/now), nitrogen was used. The ESI-MS system was set up to operate in positive ionization (PI) mode. Diagnostic fragment ions were obtained by in-source collisop-induced dissociation (CID) of the protonated molecule [M + H]^+^ after optimization of the skimmer cone voltage. Selected Ion Monitoring (SIM) was applied for scheduled analyte recording. The mobile phase solvents, which were applied for LC-ESI-MS analysis, were (A) methanol (10 mM heptafluorobutyric acid) and (B) water (10 mM heptafluorobutyric acid), respectively. The mobile phase flow was set at a flow rate of 1 mL/min. The column was kept at room temperature and the eluted analytes using an initial linear gradient program from 10% solvent A to 85% in 15 min, then going from 85% solvent A to 100% in 1 min., followed by a 100% isocratic elution of A for 3 min. An additional 10 min have been added to reach initial conditions. The injected volume, both of standard solutions and of samples, was 50 μL. The data acquisition parameters are shown in Table 3.

### 2.8. Descriptive Analysis

All measurements were conducted in triplicate. The data obtained were analyzed mathematically and graphically using Microsoft Excel (Microsoft, Redmond, DC, USA). Data were expressed as mean ± standard deviation (SD). ANOVA tests were performed, and significantly different means were compared with the Turkey’s pairwise test (*p* < 0.05) for all data collected. In addition, PCA (Principal Component Analysis) was applied to highlight a natural grouping of winemaking samples depending on their BA amounts. All the computations were performed using R-based Chemometrics Software (http://www.gruppochemiometria.it/index.php/software/19-download-the-r-based-chemometric-software, accessed on 15 July 2021).

## 3. Results and Discussion

### 3.1. Wine Samples Analysis by HPLC-UV/Vis and LC-ESI-MS

HPLC-UV/Vis and LC-ESI-MS methods were optimized for the detection of biogenic amines in red and white wine samples. Scheme of BAs isolation from wine samples is shown in Figure 2. Both for HPLC-UV/Vis and LC-ESI-MS determination, wine samples were previously filtered with 0.22 µm Millipore filters and then acidified with 10.3 M HClO_4_ and 5 mM HFBA, respectively (Section 2.5).

#### 3.1.1. Optimization and Performance Characteristics of the HPLC-UV/Vis Method

Before BA determination, the derivatization conditions were optimized as reported in Vinci et al. [38] and Vinci et al. [39]. Three different parameters have been optimized: pH, temperature, and time of reaction. The method optimization conditions are reported in Appendix A. The performance characteristics of the HPLC-UV/Vis method are shown in Appendix A. In Appendix A, a chromatographic plot of the BA standard solution is shown. Calibration curves of nine biogenic amines are shown in Appendix A.

#### 3.1.2. Optimization and Performances of the LC-ESI-MS Method

Initially, the LC-ESI-MS method was performed to investigate the fragmentation behavior of the nine BAs, based on their mass/charge ratio. In order to carry out this evaluation, standard solutions of the single column-less BAs were injected in full scan mode. It was shown that these analytes, having a low relative molecular mass, split into a very small number of fragments [38,39]. The optimized LC-ESI-MS conditions to obtain the maximum fragments are summarized in Appendix A.

To evaluate the performance of the method, linearity was taken into consideration, which was evaluated using standard solutions of the 9 BAs in 5mM HFBA acid solution [38,39]. The test results are summarized in Appendix A.

### 3.2. Biogenic Amines Determination in Wine Samples

Forty-four wine samples (24 white and 20 red) were analyzed using both HPLC-UV and LC-ESI-MS methods under the selected experimental conditions. Three replicates were performed for each determination. Table 4 shows the BA amounts and their total concentration obtained for each wine sample. By comparing the reported values of the total BA amounts, it resulted in the total concentrations of BAs being much higher in red wines than in white wines. Data, showed in Figure 3A,B, can be explained by the fact that red wines are generally less acidic than white wines, and it is known in literature that BAs are produced in high quantities at high pH [27]. Furthermore, high values of BAs in wines are not only related to high pH values but also to the complexity of the bacterial microflora. Optimal growth conditions and greater bacterial diversity are mainly observed in red wines, which, therefore, show a higher content of BAs [8]. The significant differences observed in the content of BAs reported for the samples (Table 4) are probably attributable to the fact that the presence of BAs in wines is strongly dependent on different winemaking processes, which are characterized by different pH values of wines, the duration of fermentation, the aging time, and the soil and climatic conditions under which the wines are grown [30]. The data obtained show a high PUT content in both white (nd-4.22 mg L^−1^) and red (nd-10.52 mg L^−1^) wines. In particular, it was shown that red wines also had high concentrations of HIS (nd-7.57 mg L^−1^) and TYR (nd-6.59 mg L^−1^). Although these two amines have physiological functions, their excessive intake can cause food poisoning in consumers. Furthermore, their toxic effect is enhanced by the simultaneous intake of ethanol and its catabolites present in wines. Therefore, it is essential to determine HIS and TYR simultaneously, as they present a high risk of causing toxic effects due to their vasoactive and psychoactive properties [14]. Red wines also have a higher content of TRP (nd-2.50 mg L^−1^) and SER (nd-3.80 mg L^−1^), compared to white wines which are almost absent. This is probably due to the fermentation processes to which the grapes are subjected; in fact, it has been seen that wines that also undergo malolactic fermentation have higher concentrations of BAs [27].

The analysis of variance (ANOVA) showed significant differences (*p* < 0.05) among individual and total BA values. For this reason, all the biogenic amines are considered for multivariate analysis.

PCA analysis on the samples was performed to view the dataset in a reduced size and to evaluate the data matrix to highlight natural sample grouping. Appendix A shows the loading plot of the nine amines and the total concentration of BAs in the samples, while the score plot (Appendix A) highlights the similarities and differences between the different wine samples taken into consideration. After autoscaling, two significant components were identified equal to 37.5% and 16.1% of the variance respectively for PC1 and PC2.

To better underline which biogenic amine mostly influenced the two categories of wine, the Biplot was carried out (Figure 4). It results in white wines being grouped in the negative quadrants (on the left) compared to PC1 while red wines are grouped mainly in the positive quadrants; this is explained by the fact that red wines weigh most on the presence of HIS, TYR, and PUT. This distinction that occurs in the two types of wine highlighted how red wines are the category of wine that can pose more risks to human health, as the combined presence of these BAs can lead to the risk of food poisoning. For this reason, and even though no official limit has yet been decided, some Countries, to protect the health of consumers, have established legal or recommended limits for histamine concentrations in wine [5].

## 4. Conclusions

The determination of nine biogenic amines in the white and red wine samples was carried out by applying two chromatographic methods. The LC-ESI-MS analysis offers the advantage of being a fast and reliable method for the qualitative-quantitative analysis of non-derivatized BA. This allows for identifying the presence of BA in wine samples more quickly. This is essential to quickly identify the presence of HIS and TYR in samples, since, if ingested with food, they are responsible for the main negative effects on human health (i.e., nausea, cramps, headaches, hypertension, tremors, etc.) [1]. Furthermore, the toxic effects of amines in wine can be enhanced by the synergistic effect of ethanol and acetaldehyde, which inactivates the enzymes responsible for the catabolism of BAs and increases their absorption in the gastro-intestinal wall [14]. The study highlighted the presence of all nine BA considered in the wines. Furthermore, differences in concentration were highlighted between the content of PUT, HIS, and TYR, which in red wines reached higher values, respectively of 10.52, 7.57, and 6.59 mg L^−1^, while in white wines a lower content of 4.22 was found, 4.42 and 3.71 mg L^−1^, respectively. This could probably be related to multiple factors: pH of wines, oenological processes, and hygienic conditions especially for fermentation processes, in relation to the microorganisms that are involved in alcoholic and malolactic fermentation [27,28,29,30,31,32,33,34,35,36,37,38,39]. Today, it is perhaps very difficult to obtain wines without BA, which keep all their organoleptic properties unaltered, even if one could act by controlling the critical technological factors, in particular the microorganisms involved in the fermentation processes; in this way, there would be the possibility of producing wines with low or moderate levels of BA, not dangerous for the health of consumers.

## Figures and Tables

**Figure 1 ijerph-18-10159-f001:**
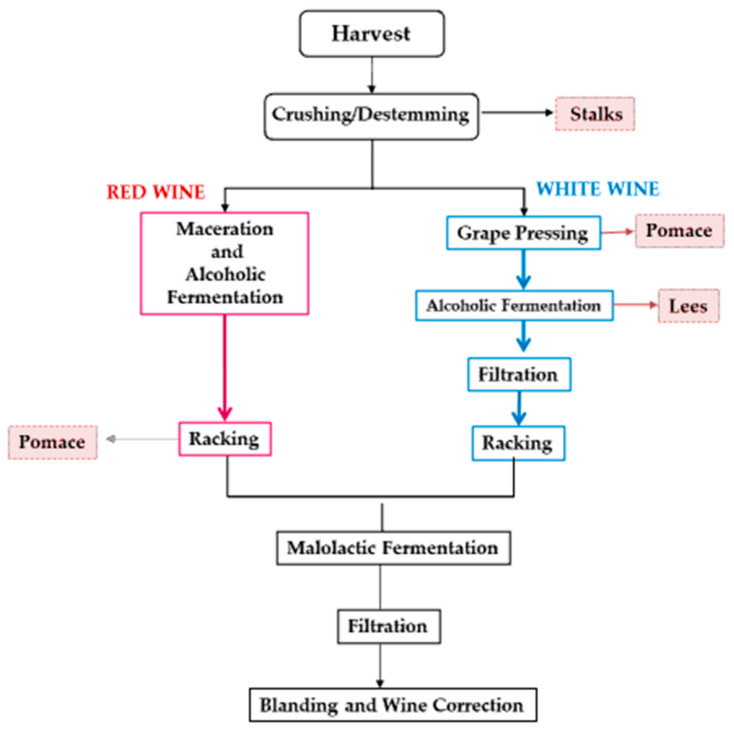
White and red vinification process.

**Figure 2 ijerph-18-10159-f002:**
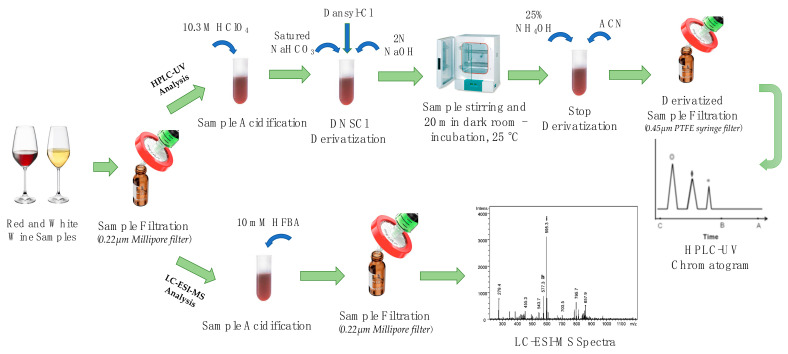
Scheme of biogenic amines determination in wine samples by HPLC-UV and LC-ESI-MS Analysis.

**Figure 3 ijerph-18-10159-f003:**
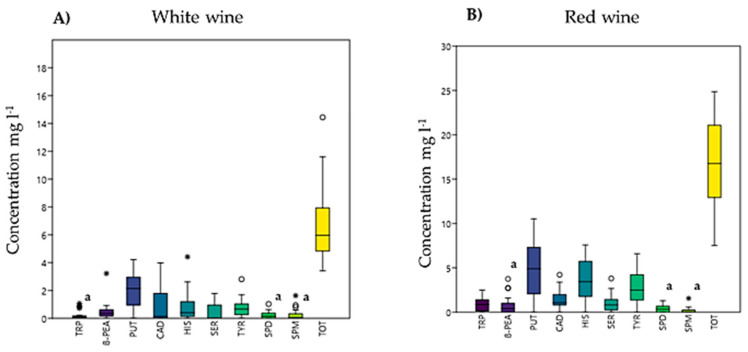
BoxPlot of Biogenic amines amount in white (**A**) and red (**B**) wine. Values are the mean sum of all samples. Bars indicate the minimum and maximum values of the BA amount. Mean values with the same letters are not significantly different according to the ANOVA test (*p* < 0.05).

**Figure 4 ijerph-18-10159-f004:**
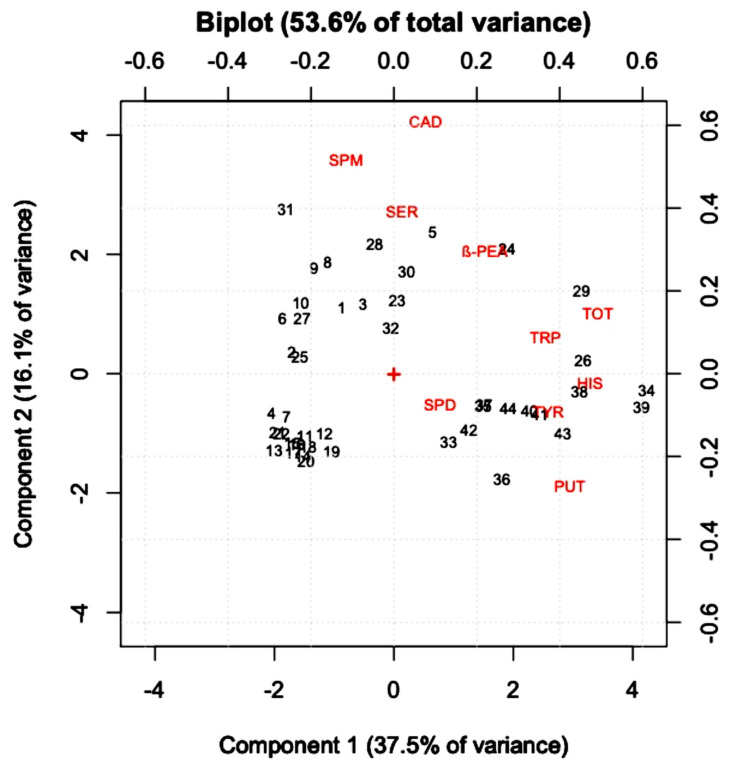
Biplot of white and red wines of Italian origin.

**Table 1 ijerph-18-10159-t001:** Physiological and pathological effects of the major BAs in food.

Biogenic Amines	Amino Acid Precursor	Physiological Effects	Pathological Effects	Ref.
Histamine	Histidine	Release of adrenaline and noradrenaline, Allergic processes, Stimulation of the smooth muscles of the uterus, intestine, and respiratory tract, Stimulation of sensory and motor neurons Control of gastric secretion	Allergic reaction (nausea, burning in the mouth, flushing of the face and body, abdominal cramps, diarrhea, swelling of the face and tongue)	[9,10,11,12,13]
Tyramine	Tryptamine	Peripheral vascularization Increase in cardiac output Increased lacrimation and salivation Increased breathing Increased blood sugar levels Noradrenaline release of the sympathetic nervous system Migraine	High blood pressure, Rapid heart rate, Tremors, Seizures, Hyperthermia	[11,14,15,16,17,18]
Putrescine	Ornithine	Hypotension Bradycardia Lockjaw Extremity paralysis	Cytotoxicity, Rule in tumors growth, Enhancement of the toxicity of other amines	[11,18,19,20]
Cadaverine	Lysine			
Tryptamine	Tryptophan	Increase in blood pressure	Relaxations, Mild euphoria, Hallucinogens	[15,21,22]
ß-Phenylethylamine	Phenylalanine	Noradrenaline release of the sympathetic nervousIncrease in blood pressure Migraine	Migraine	[11]
Spermidine	Methionine	Hypotension, Bradycardia	Acute decrease in blood pressure, Respiratory symptoms, Nephrotoxicity, carcinogenesis, tumor invasion, and metastasis, Enhancement of the toxicity of other amines	[15,23,24,25,26]
Spermine				
Serotonin	Tryptophan	Modulation of anger, aggression, mood and sexuality, appetite, Physiological homeostasis Muscle contraction Blood pressure regulation	Altered behavior and neurochemical activities, cognitive decline, muscular inflammation, and immune activation	[24,25]

**Table 2 ijerph-18-10159-t002:** Biogenic amines and microorganisms involved in wine fermentation processes.

Fermentation	Microorganism	Biogenic Amines	Ref.
Alcoholic	Spontaneous	HIS, MET, ETH, TYR, β-PEA, PUT, CAD, SPD, SPM, AGM	[8,26,27,28,29,30,31,32,33]
	*Saccharomyces cerevisiae*	AGM, ETA, ETH, PUT, TYR, CAD, β-PEA, HIS	[15,30]
	*Dekkera*/*B. bruxellensis*	ETA; MET; AGM; TRY; β-PEA; PUT; CAD; HIS; SPM	[28,29]
	*Kloeckeraapiculata*; *Candida stellata; Metschnikowiapulcherrima*	ETA; MET; AGM; TRY; β-PEA; PUT; CAD; HIS	[28]
	*Kluyveromycesthermotolerans; Schizosaccharomyces pombe V2.* *Selected S. pombe; Non-Selected S. pombe*	HIS; TYR; β-PEA; PUT; CAD	[31,32]
Malolactic	Spontaneous, *Oenococcus oeni, L. plantarum* DSM 4361; Yeast	PUT; SPD; SPM; AGM; CAD; SER; HIS; TYR; β-PEA	[8,33,34,35]
	Commercial malolactic bacteria	HIS; MET; ETH; TYR; β-PEA; PUT; CAD	[36]

HIS = Histamine; MET = Methionine; ETH = Ethylamine; TYR = Tyramine; β-PEA = β-Phenylamine; PUT = Putrescine; CAD = Cadaverine; AGM = Agmatine; ETA = Ethanolamine; TRP = Tryptamine; SPD = Spermidine; SPM = Spermine; SER = Serotonin.

**Table 3 ijerph-18-10159-t003:** Data acquisition parameters used in LC-ESI-MS for BA detection.

Biogenic Amines	MW	Channel, *m*/*z* (Relative Abundance)	Cone Voltage (V)	Retention Window (min)
Tyramine	137.2	121.2 (30), 138.3 (100)	30	0–12.85
ß-Phenylethylamine	121.2	105.1 (10), 122.3 (100)	30	12.85–16.00
Putrescine	88.2	89.3 (100)	40	0–12.85
Cadaverine	102.2	86.2 (10), 103.3 (100)	30	0–12.85
Histamine	111.1	95.2 (30), 112.1 (100)	40	0–12.85
Serotonin	176.2	160.3 (10), 177.2 (100)	40	0–12.85
Tryptamine	160.2	144.3 (40), 161.2 (100)	30	12.85–16.00
Spermidine	145.2	112.3 (10), 129.2 (10), 146.3 (100)	40	12.85–16.00
Spermine	202.3	129.2 (20), 112.3 (10), 203.4 (100)	40	12.85–16.00

**Table 4 ijerph-18-10159-t004:** Biogenic amines (mg L^−1^) amount in white and red wines.

Wine	Sample	TRP	ß-PEA	PUT	CAD	HIS	SER	TYR	SPD	SPM	Total BAs
WHITE	Wine 1	0.72 ± 0.03	0.13 ± 0.17	0.90 ± 0.13	1.54 ± 0.31	2.61 ± 0.05	0.55 ± 0.20	nd	1.03 ± 0.08	0.94 ± 0.01	9.74 ± 0.98
	Wine 2	nd	nd	1.17 ± 0.19	1.79 ± 0.19	nd	1.78 ± 0.11	0.57 ± 0.09	0.62 ± 0.16	nd	6.49 ± 0.74
	Wine 3	0.23 ± 0.02	0.16 ± 0.09	1.83 ± 0.11	2.76 ± 0.23	1.52 ± 0.07	0.96 ± 0.15	2.81 ± 0.04	nd	0.41 ± 0.04	11.59 ± 0.75
	Wine 4	nd	0.24 ± 0.01	nd	0.51 ± 0.21	nd	0.34 ± 0.02	nd	0.52 ± 0.08	nd	7.37 ± 0.32
	Wine 5	0.16 ± 0.05	3.22 ± 0.16	1.40 ± 0.12	3.97 ± 0.31	**4.42 ± 0.40**	nd	0.32 ± 0.11	0.42 ± 0.07	0.29 ± 0.02	14.44 ± 1.32
	Wine 6	nd	nd	1.70 ± 0.15	2.73 ± 0.26	0.14 ± 0.03	1.05 ± 0.13	0.26 ± 0.09	nd	0.22 ± 0.10	6.87 ± 0.76
	Wine 7	0.83 ± 0.02	nd	0.89 ± 0.12	nd	nd	0.95 ± 0.20	0.10 ± 0.03	0.21 ± 0.04	nd	5.16 ± 0.49
	Wine 8	0.89 ± 0.15	0.85 ± 0.17	0.86 ± 0.19	2.36 ± 0.24	0.51 ± 0.08	1.37 ± 0.13	0.67 ± 0.18	nd	0.58 ± 0.13	9.03 ± 1.14
	Wine 9	nd	0.92 ± 0.23	**2.57 ± 0.34**	nd	2.28 ± 0.29	1.37 ± 0.21	0.24 ± 0.09	0.26 ± 0.05	1.63 ± 0.14	9.79 ± 1.35
	Wine 10	1.07 ± 0.17	0.66 ± 0.12	0.96 ± 0.25	1.80 ± 0.12	nd	nd	0.21 ± 0.02	nd	0.80 ± 0.21	6.76 ± 0.89
	Wine 11	nd	0.88 ± 0.07	0.65 ± 0.21	nd	1.49 ± 0.16	nd	0.88 ± 0.14	0.52 ± 0.11	nd	5.02 ± 0.69
	Wine 12	nd	0.27 ± 0.04	2.95 ± 0.33	nd	1.10 ± 0.24	0.88 ± 0.12	1.04 ± 0.18	0.35 ± 0.03	nd	7.57 ± 0.94
	Wine 13	nd	0.41 ± 0.07	2.51 ± 0.27	nd	0.16 ± 0.01	nd	0.33 ± 0.09	nd	nd	3.41 ± 0.44
	Wine 14	0.02 ± 0.01	0.33 ± 0.09	**3.51 ± 0.34**	nd	0.41 ± 0.11	nd	0.85 ± 0.27	0.10 ± 0.03	0.03 ± 0.01	5.25 ± 0.86
	Wine 15	nd	0.24 ± 0.08	2.81 ± 0.39	0.32 ± 0.11	0.50 ±0.17	nd	0.67 ± 0.12	0.15 ± 0.02	0.05 ± 0.02	4.74 ± 0.91
	Wine 16	nd	0.38 ± 0.15	2.70 ± 0.41	0.30 ± 0.09	0.40 ± 0.13	nd	1.00 ± 0.09	0.08 ± 0.02	nd	4.86 ± 0.89
	Wine 17	nd	0.22 ± 0.02	3.10 ± 0.23	0.08 ± 0.03	0.22 ± 0.04	nd	0.88 ± 0.14	0.15 ± 0.05	0.05 ± 0.01	4.70 ± 0.52
	Wine 18	nd	0.60 ± 0.21	2.96 ± 0.32	nd	0.45 ± 0.10	nd	1.20 ± 0.23	0.18 ± 0.09	0.06 ± 0.02	5.45 ± 0.97
	Wine 19	0.03 ± 0.01	0.58 ± 0.18	**4.22 ± 0.39**	nd	0.50 ± 0.13	nd	1.38 ± 0.28	0.22 ± 0.06	0.08 ± 0.02	7.01 ± 0.78
	Wine 20	0.04 ± 0.01	0.15 ± 0.07	3.03 ± 0.44	nd	0.27 ± 0.03	nd	1.69 ± 0.12	0.08 ± 0.03	nd	5.26 ± 0.70
	Wine 21	nd	0.55 ± 0.11	2.41 ± 0.21	nd	0.25 ± 0.07	nd	0.34 ± 0.06	0.12 ± 0.07	0.20 ± 0.05	3.87 ± 0.57
	Wine 22	0.10 ± 0.02	0.42 ± 0.21	1.87 ± 0.19	0.20 ± 0.04	0.13 ± 0.02	nd	1.03 ± 0.17	nd	0.10 ± 0.03	3.85 ± 0.68
	Wine 23	1.28 ± 0.11	1.58 ± 0.27	2.09 ± 0.23	nd	1.85 ± 0.25	2.41 ± 0.35	1.37 ± 0.22	nd	0.51 ± 0.11	12.26 ± 1.54
	Wine 24	0.77 ± 0.21	2.75 ± 0.38	2.76 ± 0.27	4.22 ± 0.43	**3.25 ± 0.35**	nd	**3.71 ± 0.35**	0.20 ± 0.02	0.17 ± 0.05	18.82 ± 2.16
RED	Wine 25	nd	nd	1.57 ± 0.21	1.75 ± 0.19	0.51 ± 0.21	1.51 ± 0.17	0.38 ± 0.09	0.33 ± 0.05	nd	8.00 ± 0.92
	Wine 26	nd	2.68 ±0.37	**3.39 ± 0.47**	nd	**6.51 ± 0.52**	0.80 ± 0.13	**6.59 ± 0.59**	0.72 ± 0.31	0.39 ± 0.15	24.31 ± 2.54
	Wine 27	nd	nd	1.65 ± 0.12	1.90 ± 0.23	1.23 ± 0.17	2.57 ± 0.37	nd	nd	nd	8.41 ± 0.89
	Wine 28	0.47 ± 0.12	0.29 ± 0.01	2.47 ± 0.19	1.84 ± 0.19	nd	3.80 ± 0.31	**2.68 ± 0.29**	0.42 ± 0.12	0.56 ± 0.11	14.11 ± 1.34
	Wine 29	1.41 ± 0.17	3.75 ± 0.38	7.59 ± 0.56	2.91 ± 0.31	**3.10 ± 0.39**	nd	1.99 ± 0.21	nd	nd	24.25 ± 2.02
	Wine 30	1.83 ± 0.21	nd	1.52 ± 0.11	2.25 ± 0.21	1.52 ± 0.21	2.66 ± 0.19	1.20 ± 0.32	0.58 ± 0.21	0.30 ± 0.07	13.14 ± 1.53
	Wine 31	nd	0.35 ± 0.07	nd	2.80 ± 0.31	nd	0.84 ± 0.23	**2.22 ± 0.27**	0.55 ± 0.12	1.56 ± 0.23	7.51 ± 1.23
	Wine 32	0.80 ± 0.21	nd	1.96 ± 0.37	3.37 ± 0.27	**3.61 ± 0.39**	0.33 ± 0.11	0.86 ± 0.18	nd	nd	12.09 ± 1.53
	Wine 33	0.15 ± 0.13	nd	**4.42 ± 0.21**	0.83 ± 0.21	2.22 ± 0.37	nd	**5.19 ± 0.45**	1.00 ± 0.13	0.21 ± 0.04	14.02 ± 1.54
	Wine 34	2.49 ± 0.36	1.06 ± 0.31	**10.52 ± 1.23**	1.09 ± 0.19	**7.57 ± 1.05**	0.80 ± 0.24	**1.33 ± 0.16**	nd	nd	24.86 ± 3.54
	Wine 35	0.10 ± 0.02	0.34 ± 0.10	**5.88 ± 0.41**	1.10 ± 0.29	**3.25 ± 0.57**	1.40 ± 0.35	**4.28 ± 0.74**	0.77 ± 0.21	nd	17.12 ± 2.69
	Wine 36	1.05 ± 0.19	nd	**5.23 ± 1.23**	nd	**4.54 ± 0.87**	nd	**3.38 ± 0.28**	1.28 ± 0.38	nd	15.48 ± 2.95
	Wine 37	0.85 ± 0.25	0.75 ± 0.18	**6.76 ± 0.97**	0.75 ± 0.43	**4.03 ± 1.31**	0.95 ± 0.23	**2.29 ± 0.19**	nd	nd	16.38 ± 3.56
	Wine 38	1.24 ± 0.29	0.93 ± 0.23	**8.54 ± 1.25**	0.93 ± 0.27	**5.61 ± 0.98**	1.23 ± 0.24	**2.87 ± 0.24**	0.57 ± 0.15	0.08 ± 0.01	22.00 ± 3.66
	Wine 39	2.50 ± 0.12	0.18 ± 0.09	**7.22 ± 1.11**	0.81 ± 0.19	**7.11 ± 1.23**	1.00 ± 0.36	**4.61 ± 0.46**	0.33 ± 0.29	nd	23.76 ± 3.05
	Wine 40	1.64 ± 0.23	0.24 ± 0.06	**6.66 ± 1.03**	0.88 ± 0.35	**3.93 ± 0.59**	0.78 ± 0.20	**3.70 ± 0.34**	0.24 ± 0.16	0.05	18.12 ± 2.96
	Wine 41	0.57 ± 0.12	0.54 ± 0.24	**10.04 ± 1.07**	1.15 ± 0.37	**6.06 ±0.65**	0.84 ± 0.24	1.55 ± 0.37	nd	nd	20.75 ± 3.06
	Wine 42	1.33 ± 0.34	0.61 ± 0.31	**4.97 ± 0.87**	0.89 ± 0.41	2.90 ± 0.37	nd	**1.90 ± 0.23**	1.11 ± 0.14	nd	13.71 ± 2.9
	Wine 43	0.88 ± 0.27	0.47 ± 0.25	**8.32 ± 0.77**	1.05 ± 0.54	**3.80 ± 0.59**	0.54 ± 0.16	**5.05 ± 0.53**	0.66 ± 0.13	nd	20.77 ± 3.24
	Wine 44	1.42 ± 0.25	1.00 ± 0.15	**4.83 ± 0.44**	0.80 ± 0.26	**6.28 ± 0.35**	0.60 ± 0.20	**4.20 ± 0.32**	nd	nd	9.13 ± 1.97

Data were expressed how means ± standard deviation; nd = not detectable. Bolds: biogenic amines with a high concentration in the analyzed samples.

## Data Availability

Data can be accessible upon request to corresponding author (giuliana.vinci@uniroma1.it).

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
