# Peer review of "Natural Contaminants in Wines: Determination of Biogenic Amines by Chromatographic Techniques"

_ijerph, 2021, doi:10.3390/ijerph181910159_

Round 1
Reviewer 1 Report
Vinci and other authors investigated the content of biogenic amines in commercial red and white wine using two different analytical methods, including HPLC-UV and LC-ESI-MS. Based on the research, the authors claimed that red wines contained higher levels of PUT, TYR, and HIS than white wines, which is probably related to the differences in their fermentation process. This seems to be a reasonable conclusion. However, as several previous studies on the differences in biogenic amine content in red and white wine had already been published, the originality of this study is lacking.
In addition, the data obtained from method validation and wine analysis in this paper are found in other articles published by same first author as follows.
(i) Determination of biogenic amines in wines by HPLC-UV and LC-ESI-MS: a comparative study (2011).
(ii) Simple, reliable determination of biogenic amines in Italian red wines. Direct analysis of underivatized biogenic amines by LC-ESI-MS (2021).
In other word, these author re-used previous published data in this paper, which is serious violation of research ethics.
In addition, the research presented in this paper does not appear to fit the scope of this journal.
Therefore, the paper is not suitable for publication in this journal.
Other minor comments
1) Table 2: The name of microorganisms should be italicized.
2) Line 165: The volume symbol “ml” may be “µl”.
3) Figure 3B: Is it right that calibration curves are eleven, not nine? The legend of the figure presented nine biogenic amines.
4) Line 286: Please revise “hilghtlighted” to “highlighted”.
5) Table 6: For clarification, I recommend the authors to divide the table into two tables (one for red wine and the other for white wine).
Author Response
Please see the file attached.

Reviewer 2 Report
The manuscript should be accepted for publication after some minor modifications.
Table 2
At the first mention of each microorganism species I suggest to give its full Latin name (binomial) if it is possible or a genus, both written in italics; e.g. baker's yeast, Saccaromicies cervisiae is a wrong form, correct is Saccharomyces cerevisiae.
Moreover Latin names in References should be written in italics.
Table 4
The abbreviation R% (RSD) is not expanded in the captions of the Table 4 neither explained in the text.
Line 232 and 235
If I am correct, described analysis covered nine BAs, so where did eleven come from in the lines 232 and 235?
Figure 3A,B
I think there should be explained which is marked with an asterisk in the Figure 3A? Moreover there are given nine BAs in the legend of the Figure 3B, but eleven calibration curves are plotted on the graph. Is that correct?
Table 5
I am confused by the abbreviations DL (described in the captions of the Figure 4 as a detection limit) and LOD (described in the text reference to Figure 5 - line 277 - as a detection limit) used in the Table 4 and 5, respectively. I suppose both mean the same.
Table 6
I can’t find an explanation anywhere, why the columns PUT, HIS AND TYR are marked in red? Whether the BAs in wines with the highest concentration were marked in this way?
Figure 5
I suppose that the scope of the concentration scale in the graphs of the Figure 5 should be uniform.
Line 350 – 355
I think that this single sentence contains too much information which could be therefore difficult to perceive.
Author Response
Please see the file attached.

Reviewer 3 Report
There have been many studies on biogenic amines (BAs) contained in wine, but it is interesting to note that BAs in red and white wines have been qualified and quantified by HPLC and LC-MS analyses with principal component analysis. However, as a reviewer, I believe that the following points need to be revised and corrected in the manuscript.
(1) If Fig. 7 is the same as Fig. 6A and 6B combined, merge them into one.
(2) It is recommended that Table 4 and 5, and Fig. 3B and 4 be moved to Supplementary Materials.
(3) Are the concentrations of PUT, TYR, and HIS in red wine generally considered to be harmful to human health? If yes, why have humans created red wine and continued to drink it?
Author Response
Please see the file attached.

Reviewer 4 Report
The manuscript entitled “Natural Contaminants in Wines: Determination of Biogenic 2 Amines by Chromatographic Techniques” has reported the analysis of nine BA in wines by using HPLC/UV-Vis and LC-ESC-MS. The data showed that PUT, TYR, and HIS were higher concentrations in red wines. The results were interesting and the topic is of concern for the readers of International Journal of Environmental Research and Public Health. Specific comments are listed below:
Comments:
- P. 8: To optimize the operation parameters for HPLC-UV/Vis method, pH, temperature, and reaction time were considered. Please provide the chormatograms of the above conditions as supplementary information.
- P. 10: Before the analysis of wine samples. Authors demonstrated two chromatographic methods for the analysis 9 BAs, compare the advantages and disadvantages of both methods. Which one is the best selection for BA analysis? Why?
Minor:
- P. 1: line 17, “HPLC-UV/VIs” correct as “HPLC-UV/Vis”.
- P. 10: Fig. 4B, please add the caption of x and y-axis.
Author Response
Please see the file attached.

Round 2
Reviewer 4 Report
The authors have revised manuscript according to the reviewer's comments.